# Identification of a biological form in the *Anopheles stephensi* laboratory colony using the odorant-binding protein 1 intron I sequence

**Jehangir Khan**[1,2,3,4]*, **Saber Gholizadeh**[5,6], **Dongjing Zhang**[1,2,4], **Gang Wang**[1,2,4], **Yan Guo**[1,2], **Xiaoying Zheng**[1,2,4], **Zhongdao Wu**[1,2,4]*, **Yu Wu**[1,2,4]*

1 Department of Parasitology, Zhongshan School of Medicine, Sun Yat-sen University, Guangzhou, Guangdong, China, 2 Key Laboratory of Tropical Disease Control of the Ministry of Education, Zhongshan School of Medicine, Sun Yat-sen University-Michigan State University Joint Center of Vector Control for Tropical Disease, Guangzhou, Guangdong, China, 3 Department of Zoology, Abdul Wali Khan University Mardan, Khyber Pakhtunkhwa, Pakistan, 4 Chinese Atomic Energy Agency Center of Excellence on Nuclear Technology Applications for Insect Control, Sun Yat-sen University, Guangzhou, China, 5 Cellular and Molecular Research Center, Cellular and Molecular Medicine Institute, Urmia University of Medical Sciences, Urmia, Iran, 6 Medical Entomology and Vector Control Department, School of Public Health, Urmia University of Medical Sciences, Urmia, Iran

* abu_amna2013@hotmail.com (JK); wuzhd@mail.sysu.edu.cn (ZW); wuyu@mail.sysu.edu.cn (YW)

**Data Availability Statement:** All relevant data are within the manuscript and its Supporting Information files.

## Abstract

### Background

*Anopheles stephensi* Listen (1901) is a major vector of malaria in Asia and has recently been found in some regions of Africa. The *An. stepehnsi* species complex is suspected to have three sibling species: type, intermediate, and mysorensis, each with its own vector competence to the malaria parasite and ecology. To identify the members of the species complex in our *An. stephensi* insectary colony, we used the morphological features of eggs and genetic markers such as *Anste*Obp1 (*Anopheles stephensi* odorant binding protein 1), mitochondrial oxidases subunit 1 and 2 (COI and COII), and nuclear internal transcribed spacer 2 locus (ITS2).

### Methods

Eggs were collected from individual mosquitoes ($n = 50$) and counted for the number of ridges under stereomicroscope. Genomic DNA was extracted from female mosquitoes. After the amplification of partial fragments of *Anste*Obp1, COI, COII and ITS2 genes, the PCR products were purified and sequenced. Phylogenetic analysis was performed after aligning query sequences against the submitted sequences in GenBank using MEGA 7.

### Results

The range of ridges number on each egg float was 12–13 that corresponds to the mysorensis form of *An. stephensi*. The generated COI, COII and ITS2 sequences showed 100%, 99.46% and 99.29% similarity with the sequences deposited for Chinese, Indian and Iranian

**Funding:** This study is supported by the National Key R & D Program of China (no. 2020YFC1200100), National Natural Science Foundation of China (nos. 82002168 and 82072308), the 6th Nuclear Energy R & D Project (no. 20201192), and 111 project (no. B12003).

**Competing interests:** The authors have declared that no competing interests exist.

strains of *An. stephensi*, respectively. All the generated *Anste*Obp1 intron I region sequences matched 100% with the sequences deposited for *An. stephensi* sibling species C (mysorensis form) from Iran and Afghanistan.

## Conclusions

This manuscript precisely describes the morphological and molecular details of the 'var mysorensis' form of *An. stephensi* that could be exploited in elucidating its classification as well as in differentiation from other biotypes of the same or other *anopheline* species. Based on our findings, we recommend *Anste*Obp1 as a robust genetic marker for rapid and accurate discrimination (taxonomic identification) of the *An. stephensi* species complex, rather than the COI, COII, and ITS2 marker, which could only be utilized for interspecies (*Anopheles*) differentiation.

## Introduction

A wide variety of medically important insects belong to cryptic species complexes, which are morphologically identical (isomorphic), but reproductively isolated and have different seasonal prevalence, host preference, infection rates, resting habits, and biting cycles [1–3]. For instance, around 70 out of 482 species of anopheline mosquitoes act as vectors for malaria parasites and nearly 30 complexes of these have been identified so far in the world [4–6]. Because of the discovery of new biological species, the number of *Anopheles* vector species is rapidly rising [4–8]. The available information regarding the biology and distribution of *An. gambiae*, *An. culicifacies*, and *An. dirus* complexes in Africa, India, and Thailand, respectively, has demonstrated the importance of identifying the members of these species complexes [5]. Failure to discriminate among the vector and non-vector sibling species of anopheline species complexes may seriously mislead the malaria epidemiological mapping and the subsequent vector control strategies [5]. Lack of adequate knowledge about vector species complexes is playing a significant role in the current worsening scenario of human malaria in the Asian-Pacific area [9], with worldwide malaria cases rising from 217 million in 2016 to 219 million in 2017 and around 229 million in 2019 [10].

*An. stephensi* is one of the dominant malaria vectors in Middle East, the Indian subcontinent, Iran, Iraq, Bangladesh, south China, Myanmar, Thailand and Ethiopia [9,11,12]. Based on egg morphometric analysis, *An. stephensi* has three biological forms i.e. mysorensis, intermediate and type form [3] and these were identified as sibling species as Species C, Species B and Species A respectively [3,5]. The type biological form is an efficient vector of malaria in urban areas [13] whereas mysorensis is a poor vector (highly zoophilic) and limited only to rural areas, although it is susceptible to *Plasmodium vivax* (VK210B) [3,8,14]. The intermediate biological form is reported from rural and peri-urban areas with very little information about its vectorial capacity [6,15]. Despite efficient controlling strategies for malaria, *An. stephensi* is increasing in its geographic range [6]. Thus, there is a dire need for the precise identification of members of the *Anopheles stephensi* and also for the members of the other *Anopheles* complexes which is crucial in malaria surveillance, effective control, and elimination strategies [5,15].

Information regarding population genetics of *An. stephensi* is still limited [16,17]. The mitochondrial oxidases subunit 1 and 2 (COI and COII), ribosomal internal transcribed spacer 2

(rDNA-ITS2) and domain-3 (D3) loci are the common molecular markers used, but none of them have distinguished accurately the biological forms of *An. stephensi* [15,17,18]. Alternatively, intron I sequences of Odorant-binding protein 1 has been recently shown to be potential genetic marker to differentiate members of the *An. stephensi* complex [15]. Accurate identification, the spatial distribution and population dynamics of cryptic species of the *An. stepehnsi* complex has major human health implications since it directly impacts the vector control and disease management strategies [4,5].

Consequently, this study was designed with the following objectives: (i) to assess the potential of COI and COII, and ITS2 (routinely used markers) genes variations for reconstructing the phylogeny and recognition of cryptic species of *An. stephensi* in our insectary (ii) to demonstrate (as a secondary evidence) the *Anste*Obp1 intron I sequence a robust marker for rapid and accurate identification of *An. stephensi* and (iii) finally to introduce an optimized and easy protocol for sibling species identification of *An. stephensi*, based on the current molecular and morphological data.

It is indispensable to accurately characterize the insectary colony that could be used in vector control strategies such as *Wolbachia*-based, and gene drive, etc. These developing technologies are becoming more popular and important for vector population replacement/suppression, but they are highly species-specific. The preliminary sequence data (associated with mysorensis) generated through this study may contribute well to the knowledge and reliable identification (taxonomic and phylogenetics) of the mysorensis form of *An. stephensi*.

## Material and methods

### Colony maintenance

The colony of *An. stephensi* (Hor strain) has been maintained over 6 years in the insectary in 30×30×30 cm cages at Sun Yat-sen University, Guangzhou, Guangdong Province of China. Originally, this species was obtained with the courtesy of Wen-Yue Xu from Department of Pathogenic biology, Third Military Medical University, Chongqing China [19]. The rearing conditions were 28 ± 2°C, 70 ± 5% RH, and a 12:12 (L: D) h photoperiod with a 10% (W/V) sugar solution. Plastic trays (30 × 40 × 8 cm) were used for larvae rearing with deionized water and fed with IAEA 2 larval food in accordance with the standard explained procedure [20].

### Mosquito feeding, collection and morphological study of eggs

After 5–7 days of adult emergence, the female mosquitoes were allowed to feed on anesthetized white mice (Kunming strain) for 30 minutes to start egg development. After blood feeding, about 50 engorged females were randomly isolated and kept in individual properly labeled plastic tubes (50 mL) (one mosquito/tube) with a dump paper at the bottom of each tube for eggs collection. The tubes were provided with cotton soaked in 10% sugar solution. After three days, the adult females were processed further for molecular analysis. About 50 eggs were mounted on slide (each time) with a drop of water and examined under stereomicroscope with 40 × (bright field illumination) magnification to count the number of ridges on eggs (one side) as described previously [3,16]. In addition, scanning electron microscopy image was taken to clearly show the egg form.

### DNA extraction and PCR

After egg laying process, individual female mosquito from each tube was processed for DNA extraction using Dongsheng Biotech DNA extraction kit according to the manufacturer's instructions. Briefly, one mosquito was taken in 1.5 mL tube containing around 500 µL STE

buffer and a small steel ball, and homogenized (50 Hzs for 30–60 seconds). Then 5 μL of this grinding solution (for each sample) was mixed with 18 μL of DNA extraction solution in a PCR tube, mixed well and incubated for 2 minutes at room temperature. Samples were processed for PCR with thermal condition at 95˚C for 10 minutes. Afterwards, 2 μL of neutralizing fluid was added to each PCR product, mixed well and incubated for several minutes at room temperature. Finally, the extracted DNA was either kept at -20˚C or immediately processed for further amplification of target gene.

## Amplification of COI, COII, ITS2, and *AnsteObp1* fragments

PCR was performed for individual mosquito (*n* = 50) to amplify COI, COII, ITS2 and *Anste*Obp1 partial genes. The PCR reactions were carried out in a 25 μL volume and the details of used primers and PCR conditions for each marker are presented in Table 1. Double distilled H$_2$O was used as negative control instead of template DNA in PCR reactions. The PCR products were purified by using TaKaRa agarose Gel DNA extraction Kit (Japan) and the amplicons were subsequently sequenced bi-directionally (both directions) using Sanger sequencing technology.

## Sequence analysis and phylogenetic tree construction

The sequences were trimmed to remove any primer or other nucleotide contamination and double checked with Chromas software version 2.31 (www.technelysium.com.au/chromas. html). The final sequences were aligned using ClustalW [21] with the homologous sequences downloaded from GenBank and phylogenetic trees were constructed using distance Neighbor-joining and maximum likelihood Methods based on the Tamura–Nei model in MEGA7 [22].

A 120 bp fragment of *Anste*Obp1 intron I region from sequenced specimens selected from 845 bp sequenced region was used for analysis. *An. stephensi* sibling species sequences, A (KJ557463), B (KJ557452), C (KJ557455) [15] were used as representative for sequence comparisons and phylogenetic tree construction.

## Results

### Morphological analysis of the eggs

A total of 500 eggs were examined. We detected uniformity in all the observed eggs with ridges number 12–13 per egg (Fig 1). Based on the previously reported range of egg ridges, our laboratory mosquito colony was identified as the mysorensis biological form of *An. stephensi*. The previously defined criteria for identifying these three biological forms on the basis of ridges on

**Table 1. Primers detail and their respective thermal profiles in PCR amplification.**

| Primer Name | Sequence 5–3 | Size (bp) | Thermal Profile | Reference |
|---|---|---|---|---|
| COIF | TTGATTTTTTGGTCATCCAGAAGT | 877 | DT: 94˚C (4 min), 94˚C(1m), AT: 55˚C (1m) ET: 72˚C (2m), 72˚C for (7m). 32 cycles | Chavshin et al., 2014 [39] |
| COIR | TAGAGCTTAAATTCATTGCACTAATC | | | |
| COII F | ATGGCA ACATGAGCAAATT | 640 | | |
| COII R | CCACCCTTTCTGAACATTGACC | | | |
| ITS2 5.8s F | ATCACTCGGCTCGTGGATCG | 650 | DT: 95˚C (2m) 95˚C (30s), AT: 50˚C (30s), ET: 72˚C (1m), 72˚C (5m). 30 cycles | Carter et al., 2018; [40] Djadid et al., 2006 [2] |
| ITS2 28s R | ATGCTTAAATTTAGGGGGTAGTC | | | |
| *ANSTE*OBP1 F | CGTAGGTGGAATATAGGTGG | 845 | DT: 95˚C (5m) 95˚C (1m) AT: 60˚C (1.20m), ET: 72˚C for 1.20m and further 10 m. 30 cycles | Gholizadeh et al., 2018 [1] |
| *ANSTE*OBP1 R | TCGGCGTAACCATATTTGC | | | |

**F**: Forward, **R**: Reverse, **DT**: Denaturation Temperature, **AT**: Annealing Temperature, **ET**: Extension Temperature. **m**: Minute, **s**: Second.

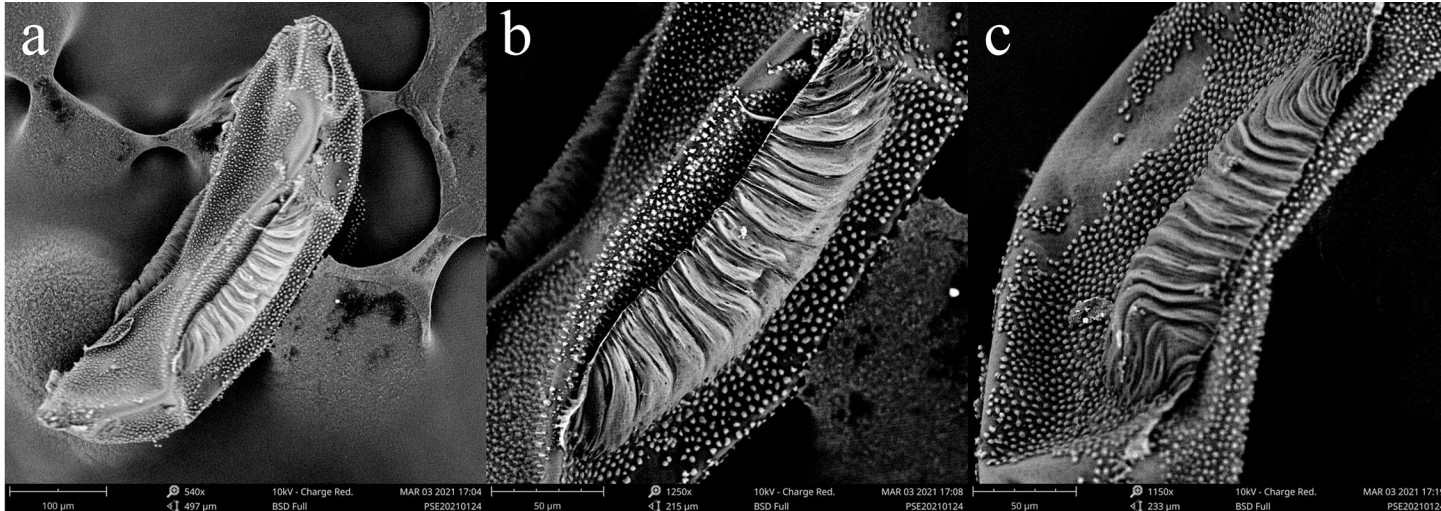

**Fig 1.** Scanning electron micrograph of *Anopheles stephensi 'mysorensis'* form egg: (a) Ventral aspect showing deck area (scale bar: 100 μm); (b) Lateral aspect showing floats and ribs (scale bar: 50 μm); (c) Lateral aspect (scale bar: 50 μm).

floating eggs are about 10–15 (mysorensis), 15–17 (intermediate) and 17–22 egg ridges (type form) [3,23].

## Molecular sequence analysis

**Phylogenetic analysis of mitochondrial oxidase subunit I (COI).**    Among 50 samples randomly selected for DNA extraction, 38 samples were sequenced; 9 samples for each of COI and COII, 7 for ITS2 while 13 samples were for *Aste*Obp1. The size of sequenced COI (MW012492 and MZ269698-MZ269705) (S1 Table) region of the lab strain was 839 bp and trimmed to 758 bp with GenBank sequences and used for analysis and phylogenetic tree construction. There are 101 sequences of *An. stephensi* COI gene submitted to GenBank, of them, seven sequences (from Iran, India, China and Brazil) were compatible with our lab strain and included in the sequence analysis (Figs 2 and S1). Multiple sequence alignment showed that the similarity between lab strain COI sequence and GenBank sequences was 99.87–100%. There were 7 mismatches in COI sequences as a transversion and 6 transitions (Figs 2 and S1). Interestingly, a sequence directly submitted from China [24] was 100% similar to our lab strain COI sequence (Fig 2). Interestingly, COI sequence of *An. stephensi* from Iran, India, Brazil and China distributed in three different clades in a phylogenetic tree (S1 Fig) (it will be because the sequences were from *An. stephensi* sensu lato). Our lab strain sequence was placed with Chinese *An. stephensi* sequence in the same clade.

**Phylogenetic analysis of mitochondrial oxidase subunit II (COII).**    The COII sequences of *An. stephensi* (*n* = 24) extracted from GenBank were compared with our lab strain sequences (n = 9). Four sequences were excluded because of shorter sequence size. COII sequences (560bp; MW431057, and MZ420723-MZ420730) (S1 Table) of lab strain showed 99.82–100% similarity within nine sequences. A limited variation was because of two mismatches in 195 (C/T), 215 (A/G) and 523 (T/A) nucleotides as transition and transversion, respectively (S2 Fig). The similarities between our sequences and 20 *An. stephensi* COII sequences available in GenBank deposited from different countries were 98.75–99.82% (Figs 3 and S2). Less than 2% variation was because of 13 mismatches as transition (*n* = 11) and transversion (*n* = 2) (Figs 3 and S2). Interestingly, mismatches in 195, 454, and 547 positions were

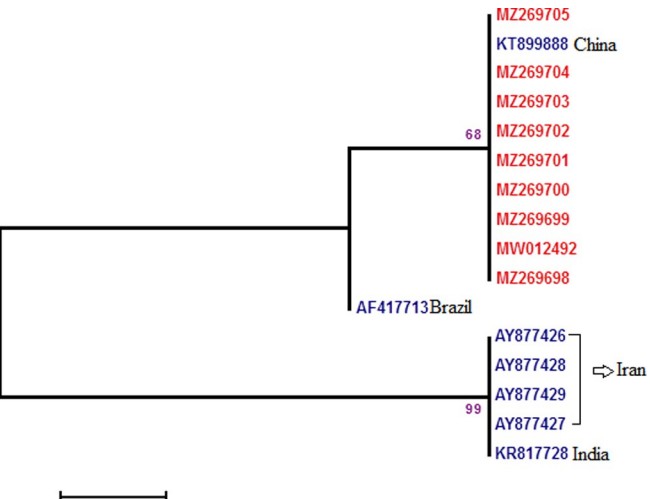

**Fig 2. Phylogeny of COI sequence from lab strain.** Bootstrap values >70 shown at nodes. Nodes without numbers had a value <70. Final ML Optimization Likelihood: -1252.592081.

specific to lab strain (S2 Fig). Phylogenetic tree constructed based on *An. stephensi* COII sequences categorized the sequences in 3 clades (Fig 3). Lab strain sequence was placed in a separate clade together with *An. stephensi* COII sequences from India and Iran (Fig 3).

**Phylogenetic analysis of nuclear internal transcribed spacer 2 locus (ITS2).** After BLAST, *An. stephensi* rDNA-ITS2 sequences were extracted from GenBank (*n* = 118). These

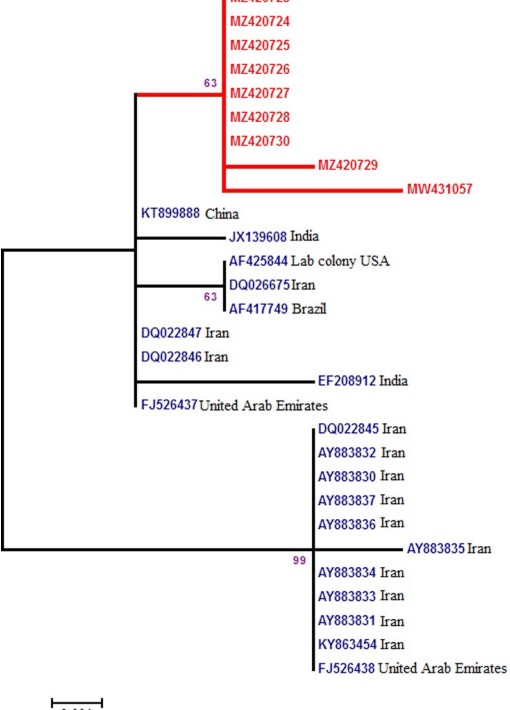

**Fig 3. Phylogeny of COII sequence from lab strain.** Bootstrap values >70 shown at nodes. Nodes without numbers had a value <70. Final ML Optimization Likelihood: -1252.592081.

sequences were previously submitted from Iran, India, Iraq, Saudi Arabia and Sri Lanka. Eleven sequences from Sri Lanka were partial and therefore, excluded from the study. Finally, 37 representative rDNA-ITS2 sequences from GenBank and sequences obtained in the current study (n = 7; MW017363 and MW017364, MZ269267- MZ269271) (470bp) (S1 Table) were used for analysis and phylogenetic tree construction (Figs 4 and S3). Comparisons of our new lab strain sequences showed 98.71% similarity with each other, while they were randomly selected from the same colony. BLAST analysis of obtained sequences showed 97.63–100% similarities with sequences reported from Iran, India, Iraq and Saudi Arabia (Fig 4). Interestingly, a sequence from India, HQ703001, showed 82.19–82.97% similarity with other rDNA-ITS2 sequences of *An. stephensi*, while its similarity with AY702482 from Iran was 98.28–99.79% (Figs 4 and S3). The similarity among Lab sequences with others was 97.63–99.57% (Fig 4). The topology of phylogenetic tree based on rDNA-ITS2 sequences of *An. stephensi* was similar to COI and COII having 3 clades with lower bootstrap values for clades (S3 Fig).

## Phylogenetic analysis of *An. stephensi* odorant binding protein 1 (*Anste*Obp1)

Multiple sequence alignment showed 100% similarity of *Anste*Obp1 intron I sequence among thirteen *An. stephensi* specimens (MW013512-MW013520 and MZ420719-MZ420722) (S1 Table). They were 100% similar with *An. stephensi* sibling C (mysorensis) (from Iran and Afghanistan), while their similarity with *An. stephensi* sibling A and B was 85% and 75.65%, respectively (Fig 5). The *Anste*Obp1intron I sequences obtained in this study were clustered in a separate clade together with *An. stephensi* sibling C (mysorensis) in a based phylogenetic tree

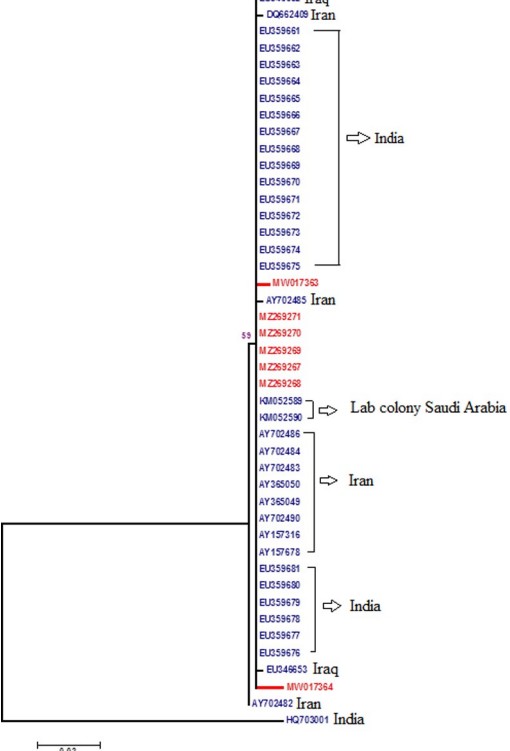

**Fig 4. Phylogeny of rDNA-ITS2 sequences from lab strain.** Bootstrap values >70 shown at nodes. Nodes without numbers had a value <70. Final ML Optimization Likelihood: -1104.708501.

```
#KJ557455_An._stephensi_C_(mysorensis)       GTGAGCTTGG GTGTCTTCTG GATATTGCTC TAATGTGTTT TTACTCTACT TGCTTTTGAC  [ 60]
#KJ557452_An._stephensi_B_(intermediate)     .....T.... .......... .......T.. .......... .CTG....TA A.T.......  [ 60]
#KJ557463_An._stephensi_A_(type)             .......... .......... .......T.. .......... .CTG....TA A.T...AA..  [ 60]
#MW013512                                    .......... .......... .......... .......... .......... ..........  [ 60]
#MW013513                                    .......... .......... .......... .......... .......... ..........  [ 60]
#MW013514                                    .......... .......... .......... .......... .......... ..........  [ 60]
#MW013515                                    .......... .......... .......... .......... .......... ..........  [ 60]
#MW013516                                    .......... .......... .......... .......... .......... ..........  [ 60]
#MW013517                                    .......... .......... .......... .......... .......... ..........  [ 60]
#MW013518                                    .......... .......... .......... .......... .......... ..........  [ 60]
#MW013519                                    .......... .......... .......... .......... .......... ..........  [ 60]
#MW013520                                    .......... .......... .......... .......... .......... ..........  [ 60]
#MZ420719                                    .......... .......... .......... .......... .......... ..........  [ 60]
#MZ420720                                    .......... .......... .......... .......... .......... ..........  [ 60]
#MZ420721                                    .......... .......... .......... .......... .......... ..........  [ 60]
#MZ420722                                    .......... .......... .......... .......... .......... ..........  [ 60]

#KJ557455_An._stephensi_C_(mysorensis)       AGAAATCTGA ATTCTGAATG TTAAATATAA TCTCCTGTCA TGCAATGTCA TCACTTTCCA  [120]
#KJ557452_An._stephensi_B_(intermediate)     .C.......G .C....C..C C....G.... .T..-----. .......... ..........  [120]
#KJ557463_An._stephensi_A_(type)             .C.......G .C....C..C C....G.... .G........ .......... ..........  [120]
#MW013512                                    .......... .......... .......... .......... .......... ..........  [120]
#MW013513                                    .......... .......... .......... .......... .......... ..........  [120]
#MW013514                                    .......... .......... .......... .......... .......... ..........  [120]
#MW013515                                    .......... .......... .......... .......... .......... ..........  [120]
#MW013516                                    .......... .......... .......... .......... .......... ..........  [120]
#MW013517                                    .......... .......... .......... .......... .......... ..........  [120]
#MW013518                                    .......... .......... .......... .......... .......... ..........  [120]
#MW013519                                    .......... .......... .......... .......... .......... ..........  [120]
#MW013520                                    .......... .......... .......... .......... .......... ..........  [120]
#MZ420719                                    .......... .......... .......... .......... .......... ..........  [120]
#MZ420720                                    .......... .......... .......... .......... .......... ..........  [120]
#MZ420721                                    .......... .......... .......... .......... .......... ..........  [120]
#MZ420722                                    .......... .......... .......... .......... .......... ..........  [120]
```

**Fig 5. Multiple sequence alignment of *Anste*Obp1 partial sequence of lab strain with three known biological forms of *An. stephensi*.**

(Fig 6). As shown previously [15,16], *An. stephensi* sibling species A and B were placed in the separate clades (Fig 6).

## Discussion

Traditionally, *An. stephensi* was classified into two races based on the number of egg ridges [23], the '*mysorensis*' and '*type*', which were considered as sub-species [25,26] or sympatric species [27,28]. Later studies used different methods to give specific taxonomic status to these three ecological/biological forms that differed in ridges number on the eggs, namely genetic crosses [3,16], cuticle hydrocarbons [29], chromosome karyotypes [30–33], spiracular index [34,35], differences in mating characteristics and behavior [5], cytogenetic properties [30,31], and molecular markers (ITS2, D3 loci, CO1 and COII) [18]. But none of these methods could clearly differentiate and give species status to the biological forms of *An. stephensi* [17,18,26].

Recently, *Anste*Obp1 is reported as a new marker for identification of the Asian main malaria vector, *An. stephensi* [15]. In the current study, we assessed the effectiveness of commonly used (COI, COII, and ITS2) and novel (*Anste*Obp1) markers, as well as morphological features (egg ridge counts), in identifying the biological form of *An. stephensi*. The extensive morphological analysis of our mosquito eggs showed that ridges number in the range 12-13/egg corresponds to the mysorensis form of *An. stephensi* [3,23]. Our results are in accordance with the previously reported range of ridges number (mysorensis) i.e. 11 to 14 [35], 13–14 [36], 10–14 [3]. Similarly, the phylogenetic analysis (Fig 6) of *Anste*Obp1 sequences of our mosquito strain showed 100% similarity with sibling species C reported from Iran and Afghanistan (the neighboring country of China). The current study has some limitations. First, the test was limited to a single laboratory colony (wild mosquitoes were not available

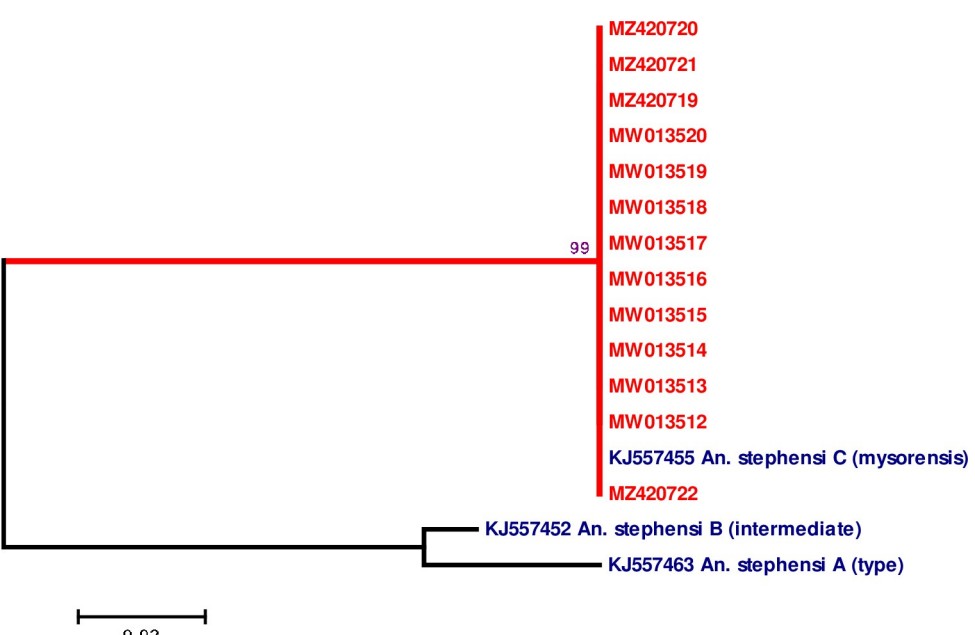

**Fig 6. Phylogenetic three constructed based on *Anste*Obp1 intron I region of *An. stephensi* sequences from Lab strain China and representative sequences from GenBank.**

because of strict vector control measures in China). Second, we have not developed a rapid PCR-based test and identification of the biological form requires sequencing, which is both labor-intensive and time-consuming. Third, only a single (the last) SNP discriminates all three forms: "C" in mysorensis, "T" in intermediate, and "G" in type. Other polymorphisms are shared between at least two forms. However, in a previous study [16], the entire wild collection of three strains of *An. stephensi* was successfully discriminated using *Anste*Obp1. This identification was attributed to the form specific/associated mutations (4–15%) within the intron region of *Anste*Obp1 between biological forms but no significant variation was noticed within the biological forms [16]. There was 99–100% similarity in the amino acids sequences of *Anste*Obp1 among these members, with a single substitution (non-synonymous) in the type form [16]. Taking together, our mysorensis associated preliminary sequence data may be exploited as representative/reference sequences for sequence comparisons and phylogenetic tree construction in similar studies in the future. Finally, our investigations, based on egg morphology and sequence analysis, endorse the use (independent) of the *Anste*Obp1 intron I sequence as a suitable molecular tool for quick and reliable identification of *An. stephensi* mysorensis form.

Vector control is fundamental for preventing the spread of malaria. Understanding population genetic structure of mosquito is imperative for shaping prevention strategies, particularly *Wolbachia*-based, gene drive, etc. The level of gene flow between mosquito populations can be predicted using population genetics research [37]. There are conflicts in the results regarding intra-species cross mating experiments. For example, reciprocal crosses between mysorensis and type strains reported a definite incompatibility [28]. Others demonstrated variations in the reproductive capacity within *An. stephensi* [38]. In contrast, no hybrid sterility was reported during a type-mysorensis cross experiments [27]. Subbarao et al (1987) did not find sterility in crossing experiments between the laboratory strains of the three ecological forms/ biological forms [3]. As a result, these experiments provide perplexing results, which should be repeated after precise recognition of members of this species complex using effective genetic

marker, such as *Anste*Obp1, and following the techniques outlined here (Table 1). Further, there is a dire need to see whether pre-mating (reproductive isolation) barriers exist among these forms in the field. Accurate identification and exploring mating compatibility/incompatibility of the wild populations with the released (lab) strain is essential for the successful operation and field application of new emerging technologies i.e. *Wolbachia*-based that has been implemented for the suppression/replacement of wild *Aedes* mosquito population in a dozens of countries to control dengue and Zika viruses [39]. If both the wild and lab (*Wolbachia* infected) mosquito populations are compatible, this approach could be used to eradicate malaria vectors in the wild.

Regarding the suitability of ITS2, COI and COII for distinguishing the biotypes of *An. stephensi*, our current observations and previous studies [17,18] confirm that these markers are not the suitable markers (based on high sequence similarity). In contrast, a new study reports both COI and COII (gene variation) as suitable markers to recognize the complexes of *An. gambiae* and *An. albitarsis* [40]. Yet our BLAST searches at GenBank database for either of ITS2, COI and COII sequences once again revealed them inappropriate for distinguishing our species strain. Our results are in accordance with [13,18,41,42]. Here, the phylogenetic analysis for COI, COII and ITS2 indicated our species 100%, 99.46% and 99.29% similar to other Chinese, Indian and Iranian strains of *An. stephensi* (Figs 2–4). Consequently, this indicates that aforesaid markers could be recommended only for identifying the species of *An. stephensi* (interspecies of *Anopheles* mosquito). Thus the independent use of *Anste*Obp1 and associated protocols mentioned in this study are recommended to be used in future investigations that involve distinguishing the members of *An. stephensi* (intra-species variation).

## Conclusion

This study finds *Anste*Obp1 as a robust genetic marker for the identification of members of the *An. stephensi* complex. We support the hypothesis based on the inability of COI, COII, and ITS2 to identify *An. stephensi*'s sibling species. This study provides important information on morphological and molecular characterization of mysorensis biological form and the associated protocols. Conducting comprehensive entomological surveillance with precise identification of sibling species of *An. stephensi* complex contribute significantly to the ongoing malaria control strategies. Consequently, we urge more research on the *An. stephensi* complex's bionomics, seasonal abundance, host and habitat preferences, plasmodium parasite susceptibility, biting cycle, and response to vector control strategies.

## Supporting information

**S1 Fig. Multiple sequence analysis of COI sequence from lab strain.** Bootstrap values >70 shown at nodes.
(DOCX)

**S2 Fig. Multiples sequence analysis of COII sequence from lab strain.** Bootstrap values >70 shown at nodes.
(DOCX)

**S3 Fig. Multiples sequence analysis of rDNA-ITS2 lab strain.** Bootstrap values >70 shown at nodes.
(DOCX)

**S1 Table. GenBank accession numbers of COI, COII, ITS2 and AnsteObp1 sequences obtained during the current study and the publicly available sequence that showed highest**

**similarity (>96% similarity) to these sequences.**
(DOCX)

## Acknowledgments

The authors thank Dr. Wen-Yue Xu from Department of Pathogenic biology, Third Military Medical University, Chongqing China, who provided us the mosquito. We also thank Mr. Hongxin Ou for his technical support. We further extend our thanks to Dr. Sarala K Subbarao, formerly Director National Institute of Malaria Research (Delhi, India) for her help in revising the manuscript.

## Author Contributions

**Conceptualization:** Jehangir Khan, Xiaoying Zheng, Yu Wu.

**Formal analysis:** Jehangir Khan, Saber Gholizadeh.

**Funding acquisition:** Zhongdao Wu, Yu Wu.

**Investigation:** Jehangir Khan, Gang Wang.

**Methodology:** Jehangir Khan, Dongjing Zhang, Yan Guo.

**Supervision:** Zhongdao Wu.

**Writing – original draft:** Jehangir Khan.

**Writing – review & editing:** Jehangir Khan, Saber Gholizadeh.

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
