## [Decision Letter · Decision Letter 0]

18 Jan 2022

PONE-D-21-36790Speculation on the possibility for introducing Anopheles stephensi as a species complex: secondary evidence based on odorant-binding protein 1 intron I sequencePLOS ONE

Dear Dr. Khan,

Thank you for submitting your manuscript to PLOS ONE. After careful consideration, we feel that it has merit but does not fully meet PLOS ONE’s publication criteria as it currently stands. Therefore, we invite you to submit a revised version of the manuscript that addresses the points raised during the review process. Please address all the comments from both reviewers. Especially include the limitations of your study as indicated by reviewer 2. Make sure that the accession numbers are working. They are not searchable now. Also, consider modifying the title as the correct title is similar to the one in the previously published paper: https://doi.org/10.1186/s12936-018-2523-y. 

We look forward to receiving your revised manuscript.

Kind regards,

Igor V. Sharakhov

Academic Editor

PLOS ONE

Journal Requirements:

( This study is supported by the National Research and Development Plan of China (No. 2016YFC1200500) and 111 project (B12003).

Prof Wu Zhongdao has received this funding.)

Reviewers' comments:

Reviewer's Responses to Questions

**Comments to the Author**

1. Is the manuscript technically sound, and do the data support the conclusions?

Reviewer #1: Partly

Reviewer #2: Partly

2. Has the statistical analysis been performed appropriately and rigorously? 

Reviewer #1: N/A

Reviewer #2: N/A

3. Have the authors made all data underlying the findings in their manuscript fully available?

Reviewer #1: Yes

Reviewer #2: No

4. Is the manuscript presented in an intelligible fashion and written in standard English?

Reviewer #1: Yes

Reviewer #2: Yes

5. Review Comments to the Author

Reviewer #1: Page 3: Authors claim "AnsteObp1 as a robust genetic marker for rapid and accurate

discrimination (taxonomic identification) of the An. stephensi species complex".

Page 11: " Finally, our investigations,

based on egg morphology and sequence analysis, endorse the use (independent) of the

AnsteObp1 intron I sequence as a new molecular tool for quick and reliable identification of all

the three biological forms of An. stephensi."

First: this claim had been proposed before on 2015, so this is not a "new" molecular tool.

Second: In this study only An. stephensi mysorensis lab strains were used. No wild anopheles and no type and intermediate forms were used.

Third: how would you use this marker to separate Type and Intermediate forms from Mysorensis?

The answer to this question is not clearly mentioned in the manuscript.

Phylogenetic trees in Fig.2-3 -4: Comparison between sequences is not clearly mentioned in the phylogenetic tree by name/region.

Reviewer #2: The manuscript by Jehangir Khan and coauthors tested the application of SNPs in the Anste Obp1 gene described in previous work for identification of biological form in the laboratory colony from China. The study is well done but it has quite a limited scope since it focuses only on one mosquito colony.

The title is not appropriate for this work as it does not introduce a new marker for species discrimination. I suggest changing the title to “Identification of a biological form in the Anopheles stephensi laboratory colony using the odorant-binding protein 1 intron I sequence.”

The authors should clearly state the limitations of their work.

First, the test was limited to a single laboratory colony. Second, the authors have not developed a rapid PCR-based test and identification of the biological form requires sequencing, which is labor- and time-consuming. Third, only a single (the last) SNP discriminate all three forms: “C” in mysorensis, “T” in intermediate, “G” in type. Other polymorphisms are shared between at least two forms.

The authors wrote that they counted for the number of ridges under stereomicroscope, however Figure 1 shows a scanning electron micrograph. Please, clarify and correct this, and also provide scale bars.

Please, correct the following:

Abstract:

An. stepehnsi species complex - change to - The An. stephensi species complex

vector competence (malaria) and ecology - change to - vector competence to the malaria parasite and

ecology.

To identify the species complex of our An. stephensi insectary colony - change to - To identify the members of the species complex in our An. stephensi insectary colony

Eggs were collected from individual mosquito - change to - Eggs were collected from individual mosquitoes

sequences in GenBank using MEGA vx. - change to - sequences in GenBank using MEGA 7.

An. stephensi sibling C - change to - An. stephensi sibling species C

Main text:

Despite efficient controlling strategies for malaria, An. stephensi is increasing in its geographic range (add reference)

chromosome karyotypes [30,31,32,33], cytogenetic properties [30,31].

Recently, AnsteObp1 is reported as a new marker for identification of the Asian main malaria vector, An. stephensi (add reference)

amino acids sequences of AnsteObp1among these members - change to - amino acids sequences of AnsteObp1 among these members

Wolbachia-based, Gene drive etc. - change to - Wolbachia-based, gene drive, etc.

Others demonstrated variations in the reproductive capacity within these biological forms of An. stephensi [34]. - change to - Others demonstrated variations in the reproductive capacity within An. stephensi [34].

Table 1: Size of ANSTEOBP1 is 900 bp, but it should be 845 bp.

6. PLOS authors have the option to publish the peer review history of their article (what does this mean?). If published, this will include your full peer review and any attached files.

Reviewer #1: No

Reviewer #2: No

---

## [Author Response · Author response to Decision Letter 0]

25 Jan 2022

Dear Editor, 

We have addressed all the points raised by the reviewers. The corrections and modifications in the manuscript have been made accordingly. Furthermore, after publication, the sequences of our manuscript will be publicly available (with working accession numbers) at NCBI.

The responses to each reviewer's individual points are listed below.

Reviewer #1: 

Page 3: Authors claim "AnsteObp1 as a robust genetic marker for rapid and accurate discrimination (taxonomic identification) of the An. stephensi species complex".

Page 11: " Finally, our investigations, based on egg morphology and sequence analysis, endorse the use (independent) of the AnsteObp1 intron I sequence as a new molecular tool for quick and reliable identification of all the three biological forms of An. stephensi." 

First: this claim had been proposed before on 2015, so this is not a "new" molecular tool.

Reply: We have replaced the word ‘new’ with ‘suitable’.

Second: In this study only An. stephensi mysorensis lab strains were used. No wild anopheles and no type and intermediate forms were used.

Reply: We have replaced the ‘three biological forms of An.stepehnsi’ with ‘An. stephensi mysorensis’.

Third: how would you use this marker to separate Type and Intermediate forms from Mysorensis? The answer to this question is not clearly mentioned in the manuscript.

Reply: This question has already been addressed in the manuscript in the discussion, such as ‘This identification was attributed to the form specific/associated mutations (4-15%) within the intron region of AnsteObp1 between biological……… [16]’. Moreover, we have cited the relevant study by Gholizadeh et al. (2015) [16] and further details may be seen by readers.

Phylogenetic trees in Fig.2-3 -4: Comparison between sequences is not clearly mentioned in the phylogenetic tree by name/region.

Reply: Comparison between sequences is now clearly mentions the phylogenetic tree by name/region.

Reviewer #2: 

The title is not appropriate for this work as it does not introduce a new marker for species discrimination. I suggest changing the title to “Identification of a biological form in the Anopheles stephensi laboratory colony using the odorant-binding protein 1 intron I sequence.”

Reply: We have changed the title as suggested.

The authors should clearly state the limitations of their work.

First, the test was limited to a single laboratory colony. Second, the authors have not developed a rapid PCR-based test and identification of the biological form requires sequencing, which is labor- and time-consuming. Third, only a single (the last) SNP discriminate all three forms: “C” in mysorensis, “T” in intermediate, “G” in type. Other polymorphisms are shared between at least two forms.

Reply: The limitations have been included now.

The authors wrote that they counted for the number of ridges under stereomicroscope, however Figure 1 shows a scanning electron micrograph. Please, clarify and correct this, and also provide scale bars.

Reply: We counted the number of ridges under stereomicroscope according to the previously published protocols (common methods) [3, 16] but we used scanning electron microscopy to present a clearer image. Now, we have added this sentence “In addition, a scanning electron microscopy image was taken to clearly show the egg form” in the concerned section. Moreover, the scale bar values are added (mentioned) now in the figure legend.

Please, correct the following:

Abstract:

1. An. stepehnsi species complex - change to - The An. stephensi species complex 

Reply: Corrected.

2. vector competence (malaria) and ecology - change to - vector competence to the malaria parasite and ecology.

Reply: Corrected.

3. To identify the species complex of our An. stephensi insectary colony - change to - To identify the members of the species complex in our An. stephensi insectary colony.

Reply: Corrected.

4. Eggs were collected from individual mosquito - change to - Eggs were collected from individual mosquitoes.

Reply: Corrected.

5. sequences in GenBank using MEGA vx. - change to - sequences in GenBank using MEGA 7. 

Reply: Corrected.

6. An. stephensi sibling C - change to - An. stephensi sibling species C

Reply: Corrected.

Main text:

1. Despite efficient controlling strategies for malaria, An. stephensi is increasing in its geographic range (add reference).

Reply: reference added.

2. chromosome karyotypes [30,31,32,33], cytogenetic properties [30,31]. Recently, AnsteObp1 is reported as a new marker for identification of the Asian main malaria vector, An. stephensi (add reference) 

Reply: Reference added.

3. amino acids sequences of AnsteObp1 among these members - change to - amino acids sequences of AnsteObp1 among these members.

Reply: Corrected.

4. Wolbachia-based, Gene drive etc. - change to - Wolbachia-based, gene drive, etc.

Reply: Corrected.

5. Others demonstrated variations in the reproductive capacity within these biological forms of An. stephensi [34]. - change to - Others demonstrated variations in the reproductive capacity within An. stephensi [34].

Reply: Corrected.

Table 1: Size of ANSTEOBP1 is 900 bp, but it should be 845 bp.

Reply: Corrected.

---

## [Editor Report · Decision Letter 1]

28 Jan 2022

Identification of a biological form in the Anopheles stephensi laboratory colony using the odorant-binding protein 1 intron I sequence

PONE-D-21-36790R1

Dear Dr. Khan,

We’re pleased to inform you that your manuscript has been judged scientifically suitable for publication and will be formally accepted for publication once it meets all outstanding technical requirements.

Kind regards,

Igor V. Sharakhov

Academic Editor

PLOS ONE
---

## [Editor Report · Acceptance letter]

10 Feb 2022

PONE-D-21-36790R1 

Identification of a biological form in the *Anopheles stephensi* laboratory colony using the odorant-binding protein 1 intron I sequence 

Dear Dr. Khan:

I'm pleased to inform you that your manuscript has been deemed suitable for publication in PLOS ONE. Congratulations! Your manuscript is now with our production department. 

Kind regards, 

on behalf of

Dr Igor V. Sharakhov 

Academic Editor

PLOS ONE